Orthostatic stability with intravenous levodopa

Siddiqi Shan H. 1
Creech Mary L. 2
Black Kevin J. 1 2 3 kevin@WUSTL.edu
1 Department of Psychiatry, Washington University School of Medicine , St. Louis, MO , USA
2 Department of Neurology, Washington University School of Medicine , St. Louis, MO , USA
3 Department of Radiology, Anatomy & Neurobiology, and Division of Biology and Biomedical Sciences, Washington University School of Medicine , St. Louis, MO , USA
Hallett Mark
Electronic publication date: 2015 Aug 27
Publication date: 2015
Volume: 3
Electronic Location ID: e1198
Received 2015 May 12; Accepted 2015 Jul 29
Copyright: © 2015 Siddiqi et al.
Copyright year: 2015
Copyright holder: Siddiqi et al.
License: This is an open access article distributed under the terms of the Creative Commons Attribution License, which permits unrestricted use, distribution, reproduction and adaptation in any medium and for any purpose provided that it is properly attributed. For attribution, the original author(s), title, publication source (PeerJ) and either DOI or URL of the article must be cited.
License URL: https://creativecommons.org/licenses/by/4.0/

Keywords: Carbidopa, Levodopa, Tourette syndrome, Heart rate, Randomized controlled trial, Blood pressure, Intravenous

Funding: U.S. National Institutes of Health R01 MH073856 K24 MH087913 M01 RR000036 P30 NS057105 C06 RR020092 UL1 RR024992 This work was funded by the U.S. National Institutes of Health (R01 MH073856, K24 MH087913, M01 RR000036, P30 NS057105, C06 RR020092, UL1 RR024992). The funders had no role in study design, data collection and analysis, decision to publish, or preparation of the manuscript.

==============================
Intravenous levodopa has been used in a multitude of research studies due to its more predictable pharmacokinetics compared to the oral form, which is used frequently as a treatment for Parkinson’s disease (PD). Levodopa is the precursor for dopamine, and intravenous dopamine would strongly affect vascular tone, but peripheral decarboxylase inhibitors are intended to block such effects. Pulse and blood pressure, with orthostatic changes, were recorded before and after intravenous levodopa or placebo—after oral carbidopa—in 13 adults with a chronic tic disorder and 16 tic-free adult control subjects. Levodopa caused no statistically or clinically significant changes in blood pressure or pulse. These data add to previous data that support the safety of i.v. levodopa when given with adequate peripheral inhibition of DOPA decarboxylase.

Introduction

The first therapeutic use of levodopa for Parkinson disease (PD) was by the intravenous route (Birkmayer & Hornykiewicz, 1961; Birkmayer & Hornykiewicz, 1998; Hornykiewicz, 2010). Oral administration is preferred clinically due to ease of use, although intravenous (i.v.) levodopa infusion has been favored in certain clinical circumstances (Chase, Engber & Mouradian, 1994; Abramsky & Goldschmidt, 1974; Horai et al., 2002; Mizuno et al., 2009).

The i.v. route has advantages for some research purposes as well (Black et al., 2004; Black et al., 2010; Black et al., 2015). However, levodopa is approved by the U.S. Food and Drug Administration for treatment of PD and other parkinsonian conditions only in a tablet formulation. In the U.S., giving an approved drug by another route for research purposes may require an investigational new drug (IND) application if changing the route of administration “significantly increases the risks (or decreases the acceptability of the risks)” (§21 CFR 312.2(b)(iii): http://www.accessdata.fda.gov/scripts/cdrh/cfdocs/cfcfr/CFRSearch.cfm?fr=312.2). Some have assumed that this might hold for i.v. compared to oral levodopa.

In fact, however, numerous studies have reported on brief (<24 h) infusions or large single-dose i.v. boluses, and i.v. levodopa has been tolerated approximately as well as oral levodopa (Abraham et al., 2015; Siddiqi et al., 2015). One study even deliberately attempted to induce hallucinations by giving high-dose i.v. levodopa to patients at high risk, but produced no hallucinations (Goetz et al., 1998). However, data on hemodynamic effects of i.v. levodopa have been limited over the past 20 years, and such results have not yet been quantitatively reported in the presence of a peripheral decarboxylase inhibitor (Abraham et al., 2015; Siddiqi et al., 2015).

Here we provide quantitative data, from a double-blind, random-allocation crossover study, on orthostatic blood pressure and pulse responses to i.v. levodopa in the presence of adequate carbidopa pretreatment.

Methods

Subjects

Forty-four generally healthy adults (23 with Tourette syndrome or chronic motor tic disorder [TS] and 21 tic-free healthy control [HC] subjects) enrolled in a study investigating dopaminergic effects on cortical function during a working memory task as measured by functional magnetic resonance imaging (ClinicalTrials.gov identifier NCT00634556). The study was approved by the Human Research Protection Office (IRB) of Washington University in St. Louis (project #05–0832, #201105100), and all subjects provided written documentation of informed consent prior to participation. This study was performed under FDA IND #69,745, Kevin J. Black, Sponsor-Investigator. After the first 10 subjects had been enrolled, the FDA asked us to record orthostatic blood pressure and pulse before and during the infusions. No subjects took any dopaminergic or antidopaminergic medications at baseline, including levodopa, dopamine agonists, or antipsychotics. Six subjects dropped out or were withdrawn from the imaging study (claustrophobia 1, abnormal structural MRI 1, scheduling problems 2, vomiting 2). Complete vital signs were still available for one of the two subjects who were excluded from the imaging study due to vomiting, so these data were included. Therefore, 15 of the 44 subjects were not included in vital signs analysis (10 subjects enrolled before initiation of vital sign collection, 5 subjects dropped out before full range of vital signs could be collected), leaving 29 subjects for analysis.

Medications

No subjects were taking dopaminergic or antidopaminergic medications at baseline. All subjects avoided caffeine, nicotine, and proteins starting at midnight before the morning of the study. Water, juice, and other non-protein food items were allowed prior to study initiation, but subjects had no oral intake during the study period. At least 1 h after taking 200 mg carbidopa by mouth, levodopa was given intravenously in a 2 mg/mL solution according to the “final protocol” described in Black et al. (2003). Specifically, a loading dose of 0.6426 mg/kg was given i.v. over 10 min followed by a maintenance infusion at 2.882 × 10−5 mg/kg/min × (140 yr-age)/yr for an additional 90 min. A 35-year-old patient weighing 70 kg would receive a total dose of 64 mg (in 32 mL of normal saline); an oral dose of 150–200 mg would provide the same total absorbed dose, though over a slower time scale (Sasahara et al., 1980; Robertson et al., 1989; Kompoliti et al., 2002). The mean peak levodopa plasma concentration (Cp) with this method was ∼2,350 ng/ml, and the steady-state Cp was ∼600 ng/ml (Black et al., 2003). On a separate day at least 1 week later, each patient received a placebo infusion after carbidopa. The order of the levodopa and placebo infusions was balanced. Both placebo and levodopa infusions were initiated between 8:00 AM and 10:00 AM on the study day. Study staff were blinded to the order of infusions, so both infusions occurred at approximately the same time of day.

Measurements

Vital signs were measured for each patient before and after the infusion on each day. Each vital sign measurement included heart rate, systolic blood pressure, and diastolic blood pressure (P, SBP, DBP) measured when the subject had lain supine for at least 5 min, and repeated after at least 1 min standing, a protocol that was approved by the FDA reviewer. The method of blood pressure measurement was consistent between pre-infusion and post-infusion measurements: most measurements were taken electronically, but measurements by manual sphygmomanometry were conducted on a minority of study days. At the end of the study, these measurements were available on both infusion days for 29 patients. Subjects and staff measuring vital signs were blind to infusion order.

After each infusion, subjects completed the Pittsburgh Side Effects Rating Scale (Pelham Jr, 1993), a self-rated checklist of common psychotropic side effects in which scores range between 0 and 57. Before and after each infusion, subjects completed a visual analog scale (on a scale of 1–100) to rate light-headedness–dizziness, nausea–vomiting, sleepiness, and overall health.

Statistical analysis

Means for each vital sign parameter were compared between levodopa treatment days and placebo treatment days by paired t tests. For each parameter, within-subjects confidence intervals were computed based on patients for whom the parameter was available for both placebo and levodopa days (Morey, 2008). Pittsburgh Side Effects Rating Scale scores and change in visual analog scale scores for adverse effects were compared similarly.

Results

Fourteen patients received placebo infusion on the first study day and levodopa on the second day, while 15 received the converse. All data were collected as intended with the exception of a standing pulse for one subject and post-infusion standing blood pressure for one patient after levodopa infusion and for one patient after placebo infusion. Baseline clinical characteristics are summarized in Table 1.

Table 1 Baseline clinical characteristics of subjects.

Age (years)	32.7 ± 11.2	
Weight (kg)	79.1 ± 12.4	
Sex	21 M, 8 F	
Concurrent antihypertensives	10.3%*	
Concurrent dopaminergic medications	0%	
Notes.

* Of the three patients taking antihypertensives, one was for hypertension; the other two were taking centrally acting α2-adrenergic agonists for treatment of Tourette’s syndrome.

No significant difference was found between vital sign parameters during levodopa versus placebo infusions (Table 2, Fig. 1). Standing increased P and DBP, and the magnitude of this change increased somewhat from earlier to later in the day, but none of these changes differed between levodopa and placebo. The largest absolute orthostatic increases in P, both of which were found on the post-infusion measurements, were 11.7 bpm on the placebo day and 12.3 bpm on the levodopa day. The largest absolute orthostatic increases in DBP, also found on the post-infusion measurements, were 7.4 mmHg in the placebo group and 2.0 mmHg in the levodopa group (p = 0.20). For the differences between levodopa and placebo for all vital sign parameters (supine P/SBP/DBP, standing P/SBP/DBP, orthostatic change in P/SBP/DBP), paired p values ranged between 0.16 (for supine SBP) and 0.92 (for standing SBP). Additionally, no significant difference was found for adverse effect scales (Table 3).

Figure 1 Orthostatic vital signs before and after levodopa infusion.

No significant changes were observed between IV levodopa or placebo days in (A) heart rate, (B) systolic blood pressure, or (C) diastolic blood pressure. Values shown are mean ± S.D. for all data. (See Table 2 for means and 95% confidence intervals from the paired analysis.)

Table 2 Vital sign parameters after levodopa and placebo infusions.

Means and 95% confidence intervals are shown for individual vital sign parameters after levodopa and placebo infusions, for subjects who had data on both days. No significant levodopa-placebo difference was found in any parameter. Units for BP: mmHg; units for P: beats per minute.

Condition	Number of pairs	Levodopa (95% CI)	Placebo (95% CI)	p	
SBP (supine)	29	118 (115, 122)	121 (118, 125)	0.16	
SBP (standing)	27	117 (112, 122)	120 (115, 125)	0.19	
SBP (orthostatic change)	27	−1 (−6, 4)	−1 (−6, 4)	0.91	
DBP (supine)	29	75 (72, 78)	74 (71, 77)	0.86	
DBP (standing)	27	77 (72, 82)	82 (77, 87)	0.18	
DBP (orthostatic change)	27	2 (−2, 7)	7 (2, 12)	0.20	
P (supine)	29	62 (61, 64)	61 (59, 63)	0.34	
P (standing)	28	75 (72, 78)	73 (70, 76)	0.35	
P (orthostatic change)	28	13 (10, 16)	12 (9, 15)	0.65	

Table 3 Self-reported side effects.

Mean and 95% confidence intervals are shown for self-reported side effects with levodopa and placebo infusions.

Parameter	Placebo (95% CI)	Levodopa (95% CI)	p	
Pittsburgh side effects rating scale	2.0 (1.2, 2.9)	3.0 (1.4, 3.8)	0.25	
Change in VAS (light-headedeness)	0.0 (−2.1, 2.1)	3.0 (−1.1, 7.1)	0.26	
Change in VAS (nausea)	0.7 (−0.8, 2.3)	−0.2 (−3.7, 3.4)	0.67	
Change in VAS (sleepiness)	2.8 (−3.1, 8.6)	4.9 (−0.9, 10.7)	0.88	
Change in VAS (overall health)	−1.8 (−6.2, 2.6)	−1.7 (−8.6, 5.2)	0.88	

Discussion

These data in generally healthy young adults further support more comprehensive data from previous studies suggesting that i.v. levodopa, at a dose that produces biologically meaningful effects on parkinsonism, does not meaningfully affect orthostatic vital signs when it is given after adequate inhibition of DOPA decarboxylase. Previous studies supporting this conclusion are reviewed elsewhere (Abraham et al., 2015), but here we summarize the key data.

Even before the advent of peripheral decarboxylase inhibitors, large doses of i.v. levodopa were observed to have minimal effects on blood pressure. Moorthy et al. (1972) gave 100–200 mg levodopa i.v. over 10 min to 8 cardiac patients ages 40–77, and reported slight increase in heart rate, aortic and pulmonary arterial pressures, cardiac output, oxygen consumption, heart rate, and systolic and diastolic aortic and pulmonary arterial pressure, along with a slight decrease in systemic arterial resistance; however, specific data and statistical significances were not reported and there was no placebo group. Of note, all of these parameters recovered 30 min after infusion and no subjective symptoms were reported, so the authors concluded that intravenous levodopa was safe even in PD patients with advanced cardiovascular disease (Moorthy et al., 1972). Bruno & Brigida (1965) and Baldy-Moulinier et al. (1977) monitored pulse and BP at frequent intervals after i.v. infusions of 2 mg/kg (140 mg) over 5 min and 125 mg in 15 min respectively, but provided no quantitative data; the authors simply state that there was no change in arterial BP (Baldy-Moulinier et al., 1977, p. 184) or that there were no significant clinical problems with the infusion (Bruno & Brigida, 1965).

The non-neurological side effects of levodopa are further prevented or ameliorated by carbidopa, a peripheral decarboxylase inhibitor whose purpose is to prevent levodopa from being converted to dopamine in the peripheral circulation (Barbeau & Roy, 1976; Cotzias, Papavasiliou & Gellene, 1969). Peripheral decarboxylase inhibitors revolutionized the treatment of PD 45 years ago (Papavasiliou et al., 1972) by reducing autonomic and gastrointestinal effects of oral levodopa, most commonly dose-related nausea, dizziness or orthostatic hypotension. Smaller doses of peripheral decarboxylase inhibitors (25–50 mg of carbidopa) have minimal impact on the autonomic effects of intravenous levodopa (Irwin et al., 1992), but the same occurs with oral levodopa after 50 mg of benserazide (Noack et al., 2014), suggesting that the route of administration of leovodopa is not the key difference.

By contrast, here a larger dose of carbidopa (200 mg), given early enough that adequate absorption could occur before levodopa administration, effectively prevented any autonomic effects. One might posit that these favorable results are due to the younger sample without Parkinson disease, because hypotension has been observed with levodopa (without PDIs) in several PD studies (Whitsett & Goldberg, 1972; Calne et al., 1970; Sénard et al., 1995; Haapaniemi et al., 2000; Bouhaddi et al., 2004; Wolf et al., 2006). However, hypotension with levodopa in PD is also suppressed by larger doses of PDIs (Mehagnoul-Schipper et al., 2001); similarly, with carbidopa dosed as in the present study oral levodopa produced no mean change in cerebral blood flow (Hershey et al., 1998). Finally, a different brain imaging study in PD that dosed carbidopa similarly found no significant differences in BP or P after vs. before i.v. levodopa (Black et al., 2010, and K Black, 2010, unpublished data). Thus we ascribe the positive results in the present study to the larger and earlier dosing of carbidopa.

Sedation is the most common central side effect of levodopa, and patients with advanced PD also may experience dyskinesias, hallucinations, or confusion. More recently, attention has also been given to gambling, paraphilias, and other disinhibited behavior that emerges in a substantial minority of patients treated with dopamimetics (Black & Friedman, 2006), but these complications are more common with synthetic dopamine agonists and have been reported only after chronic treatment (Weintraub et al., 2010; Poletti et al., 2013). We are unaware of any evidence that central side effects are more common with intravenous levodopa than with oral levodopa.

Overall, these data further elucidate the safety profile of intravenous levodopa and reaffirm that it causes no meaningful change in orthostatic vital signs when combined with oral carbidopa at the dose and schedule used in the present study. This adds to prior data showing an overall safety profile comparable to that of oral levodopa, and may help alleviate some of the regulatory concerns regarding the use of intravenous levodopa in research.

Supplemental Information

Supplemental Information 1 Data: vital signs and subject characteristics

See “legend” tab for a description of variables.

Click here for additional data file.

Supplemental Information 2 Informed consent document

Click here for additional data file.

The authors thank the Tourette Syndrome Association and its Greater Missouri chapter for help with recruitment. Tamara Hershey, Meghan C. Campbell, Elda Shipley, Jonathan Koller, Samantha Ranck, Kathryn Vehe, and Gary Queensen contributed to the overall study from which the data reported here were drawn.

Additional Information and Declarations

Competing Interests

Author Contributions

Human Ethics

Dr. Black is an Academic Editor for PeerJ.

Shan H. Siddiqi analyzed the data, wrote the paper, prepared figures and/or tables, reviewed drafts of the paper.

Mary L. Creech performed the experiments, reviewed drafts of the paper.

Kevin J. Black conceived and designed the experiments, performed the experiments, analyzed the data, wrote the paper, prepared figures and/or tables, reviewed drafts of the paper.

The following information was supplied relating to ethical approvals (i.e., approving body and any reference numbers):

The study was approved by the Human Research Protection Office (IRB) of Washington University in St. Louis (project #05–0832, #201105100), and all subjects provided written documentation of informed consent prior to participation. This study was performed under FDA IND #69,745, Kevin J. Black, Sponsor-Investigator.

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
