# Peer review of "Orthostatic stability with intravenous levodopa"

_PeerJ, doi:10.7717/peerj.1198_

## Round 0.1 · original submission · Major Revisions

· Academic Editor

Major Revisions

This paper has some worthwhile data to report. Revision is necessary to improve the detail of the method and results, and consider some alternative views in the Discussion.

·

Basic reporting

There are areas where the article fails to meet standards:
-The subjects are not described. Demographics including age, gender, history of hypertension, other use of dopaminergic or cardiovascular medications should be included.
-Methods should include when orthostatic blood pressures were taken (immediately, 1 minute, 2 minutes after standing?). Were both placebo and levodopa infusions done at the same time of the day? Were patients fasting?
-Table 1 should include the N for each measurement rather than the total. The data was analyzed as paired thus it is misleading to report the total for each measurement and should be reported as pairs or if unequal then the numbers for each given.
-p values in the table would be better reported than giving the range of p values.
-Patient's symptoms were not included. To make this clinically meaningful, presence of orthostatic symptoms would be useful.

Experimental design

This is a convenience sample.
-How were blood pressures recorded? Manually or electronically? Was it the same examiner?
-What was the volume of the infusion? Increasing volume through a bolus and 90 minute infusion may account for what appears to be an increase in supine diastolic BP after infusion and may negate influences of levodopa. This needs to be addressed.

Validity of the findings

-First sentence in discussion is overstated. One cannot say that there is no meaningful effect on cardiovascular system when only orthostatic blood pressure and pulse are reported.
-Conclusions should acknowledge the patient population studied.
-Authors should acknowledge that larger studies may find a difference as the diastolic blood pressure change

Additional comments

Overall it is good to report these results. There are some issues that could be addressed to improve the manuscript.

Reviewer 2 ·

Basic reporting

Adequate.

Experimental design

Double-blind, crossover comparison of levodopa infusions and placebo infusions. Appropriate.

Validity of the findings

Finds are clear cut. It is the interpretation of the results that can be questioned.

Additional comments

This manuscript describes a crossover study of pulse and blood pressure supine and standing after 200 mg of oral carbidopa and a placebo or levodopa infusion with a loading dose and maintenance infusion for 100 minutes in subjects with motor tics and in normal controls. There were no significant differences between pulse and blood pressure recordings for the placebo and levodopa infusions although the levodopa infusion produced plasma concentrations of levodopa that would be therapeutic in PD.
There is no quibbling with the results of the study.
The conclusions of the authors are, however, questionable. The authors’ conclusion is that “i.v. levodopa, at a dose that produces biologically meaningful effects on parkinsonism, does not meaningfully affect the cardiovascular system when it is given after adequate inhibition of DOPA decarboxylase.” This is based on the assumption that all vascular effects of levodopa are peripheral and will be blocked with a large dose of carbidopa. There is evidence that levodopa may have central effects on blood pressure in PD (Irwin, R.P. 1992) and certainly hypotension is a recognized side effect of oral and levodopa infusions in PD subjects.
Citing an abstract (Abraham) and a publication in preparation (Siddiqi) is a questionable practice.

Reviewer 3 ·

Basic reporting

Dear Editors

The authors report on the impact of an intravenous levodopa challenge on cardiovascular parameters in a group of cardiovascular healthy individuals (patients suffering from Tourette syndrome and healthy controls). Their primary hypothesis is that i.v. levodopa would not affect blood pressure and heart rate significantly, when the peripheral action of the compound is blocked sufficiently with an aromatic L-amino acid decarboxylase (AADC) such as carbidopa. According to the reported data (result section and figure 1), there was no significant (group wise) difference in systolic and diastolic blood pressure as well as heart rate between the three measurements: baseline, placebo, and i.v. levodopa. Baseline measurements (one week apart) before placebo and levodopa treatment did also not differ significantly.

Experimental design

see point 3 below

Validity of the findings

Major concerns:
1. The manuscript lacks a description of the participants. Please provide data on gender, age, BMI and clinical history (if any) concerning diseases potentially affecting the cardiovascular system (diabetes etc.).
2. With respect to the patients suffering Tourette syndrome; were the patients on or of concomitant drugs?
3. I do recognize that the reported trial was not primarily designed to reveal cardiovascular effects of i.v. levodopa. But before concluding that levodopa does not produce meaningful cardiovascular effects (discussion; first sentence), I wonder if the study is sufficiently powered to detect minor differences (power calculation missing)?
4. The authors need to describe the methods used to assess blood pressure in more detail; length of baseline (a three minute supine period would be too short), assessment of blood pressure performed automated or manually by sphygmomanometry or “intra-arterial line”; how many BP recordings were obtained to defined “baseline”
5. Levodopa, especially when given to drug naïve individuals and when administered with an AADC to increase its central effect, usually induces side effects such as nausea. Did the study participants suffer comparable side effects, that in turn might have confounded the cardiovascular parameters?
6. Orthostatic hypotension is a common side effect of oral levodopa+aadc (discussion, line 118). There are several studies reporting on a vasodepressor effect of oral levodopa (1-6). In an own study (7), oral administration of 200 mg levodopa + 50 mg of an aadc (benserazide) induced a significant reduction in mean arterial blood pressure (-15%, p < 0.001) and cardiac stroke volume (-13%, p < 0.01). One could argue that autonomic involvement in Parkinson’s disease unveils minor cardiovascular effects of (oral) levodopa+aadc. Otherwise the authors would need to discuss the different actions of levodopa depending of the route of administration (oral versus i.v.).

Minor concerns
7. please correct the ordinate label in figure 1a: bpm instead of mmHg


[1] Whitsett TL, Goldberg LI. Effects of levodopa on systolic preejection period, blood pressure, and heart rate during acute and chronic treatment of Parkinson's disease. Circulation 1972;1:97-106.
[2] Calne DB, Brennan J, Spiers AS, Stern GM. Hypotension caused by L-dopa. Br Med J 1970;5694:474-5.
[3] Senard JM, Verwaerde P, Rascol O, Montastruc JL. Effects of acute levodopa administration on blood pressure and heart variability in never treated parkinsonians. Hypertens Res 1995;S175-S177.
[4] Haapaniemi TH, Kallio MA, Korpelainen JT, Suominen K, Tolonen U, Sotaniemi KA, et al. Levodopa, bromocriptine and selegiline modify cardiovascular responses in Parkinson's disease. J Neurol 2000;11:868-74.
[5] Bouhaddi M, Vuillier F, Fortrat JO, Cappelle S, Henriet MT, Rumbach L, et al. Impaired cardiovascular autonomic control in newly and long-term-treated patients with Parkinson's disease: involvement of L-dopa therapy. Auton Neurosci 2004;1-2:30-8.
[6] Wolf JP, Bouhaddi M, Louisy F, Mikehiev A, Mourot L, Cappelle S, et al. Side-effects of L-dopa on venous tone in Parkinson's disease: a leg-weighing assessment. Clin Sci (Lond) 2006;3:369-77.
[7] Noack et al. Parkinsonism Relat Disord. 2014;20(8):815-8

---

## Round 0.2 · accepted · Accept

· Academic Editor

Accept

The revision is adequate and addresses all the concerns of the three reviewers.